# Evaluation of the CRTM Land Emissivity Model over Grass and Sand Surfaces Using Ground-Based Measurements

Yidan Wang [1,2], Wenying He [1,3,*], Minzheng Duan [1,3], Hailei Liu [2], Hongbin Chen [1,3], Congzhen Han [1,3] and Weidong Nan [1,4]

1 Key Laboratory of Middle Atmosphere and Global Environment Observation, Institute of Atmospheric Physics, Chinese Academy of Sciences, Beijing 100029, China; 3210307037@stu.cuit.edu.cn (Y.W.); dmz@mail.iap.ac.cn (M.D.); chb@mail.iap.ac.cn (H.C.); c.han@mail.iap.ac.cn (C.H.); nanweidong@mail.iap.ac.cn (W.N.)
2 College of Electronic Engineering, Chengdu University of Information Technology, Chengdu 610225, China; liuhailei@cuit.edu.cn
3 University of Chinese Academy of Sciences, Beijing 100049, China
4 Xianghe Observatory of Whole Atmosphere, Institute of Atmospheric Physics, Chinese Academy of Sciences, Langfang 065400, China
* Correspondence: hwy@mail.iap.ac.cn

**Abstract:** Microwave surface emissivity is complex and variable, leading to increased difficulty in accurately retrieving atmospheric parameters and assimilating satellite microwave observations over land. The Community Radiative Transfer Model (CRTM) land emissivity model is a useful tool for providing microwave emissivity over complex surfaces. By combining the model with ground measurements from a mobile multi-surface observing system at the Xianghe site, China, the performance of the land emissivity model is evaluated over grass and sand surfaces. The simulated and measured emissivity agrees at both polarizations over the grassland surface but a more significant difference is observed at the horizontal polarization over the sand surface. To solve this problem, the *Q/H* module for soil reflectance roughness correction in the CRTM emissivity model was replaced with the $Q_p$ module for the sand surface. This results in a significant improvement in the horizontal polarization simulation, with the corresponding mean bias error (MBE) reducing from 0.08 in the *Q/H* module to less than 0.03. The adjustment demonstrates that the $Q_p$ module more effectively corrects the roughness effect on horizontally polarized emissivity for bare soil surfaces. For grassland, the CRTM emissivity model with the *Q/H* module demonstrates accurate simulations, showing its suitability for vegetated land surfaces.

**Keywords:** microwave land surface emissivity; CRTM emissivity model; ground-based measurements; reflectance roughness correction





## 1. Introduction

Due to its long wavelength, microwave signals can penetrate clouds and surface, and are easily affected by complex surface features over land, such as surface type, vegetation, and surface roughness. This makes surface microwave emissivity over land more complex and difficult to properly describe than on the ocean surface. On the other hand, the higher (~0.90) surface microwave emissivity produces a strong surface radiance signal, which is mixed with the atmospheric radiance signal detected using space-borne microwave equipment, thereby significantly affecting the accuracy of atmospheric parameters retrievals, such as cloud and precipitation, over land. In short, microwave surface emissivity is complex and variable, resulting in increasing difficulty in accurately retrieving atmospheric parameters and assimilating satellite microwave observations into numerical models over land [1–3]; therefore, it is crucial to provide accurate land surface emissivity [4].

Currently, there are three main methods for obtaining land surface emissivity: (1) field observation experiments, (2) emissivity models, and (3) retrieval from satellite observations. Field observation experiments involve scanning observations using ground-based microwave radiometers over different surface types. The variations in emissivity with surface type and scan angle of the radiometer can be derived through experimental observations. Field experiments under controlled conditions can provide high temporal resolution of surface emissivity data, allowing detailed analysis of the impact of surface processes on emissivity [5–7].

Based on experimental observations and the physical principles of surface radiation, emissivity models have been developed [8–12]. Due to limited conditions for field experiments, such as the range in frequencies and surface types, the developed emissivity models are only applicable in certain situations. For instance, the emissivity model established by Wang et al. [13] is suitable for bare soil at lower frequencies. The emissivity model developed by Isaacs et al. [14] is fit for vegetation canopies, using a radiative transfer model with a large number of canopy optical parameters. In addition, these emissivity models require many detailed surface parameters as the input, such as surface temperature, soil temperature and moisture, vegetation type, and so on. Due to the many uncertainties associated with these complex parameters, it remains a challenge to accurately obtain emissivity using emissivity models globally.

In recent years, with the rapid development of satellites, space-borne microwave observations with wide coverage have been used to estimate the regional and global distributions of surface microwave emissivity [15–19]. In order to avoid the influence of clouds and precipitation, most of the surface microwave emissivity retrieved from satellite observations can only be used for clear sky conditions.

Although using microwave emissivity models or satellite measurements can directly obtain land surface microwave emissivity, it is a complex process that combines plentiful auxiliary data to fulfill the calculation. Therefore, the accuracy of surface microwave emissivity obtained from both emissivity models and satellite observations needs further validation with in situ observations. A ground-based observation system developed by our group [20] has accumulated a large amount of in situ measurements of microwave emissivity on several typical surfaces in recent years; this provides favorable conditions for validating the emissivity model.

The land emissivity model, developed by Weng et al. [8], has been widely used in the Community Radiative Transfer Model (CRTM) for radiance assimilation in numerical weather prediction models. Prigent et al. [21] evaluated and compared the CRTM emissivity model with the global land surface emissivity calculated by TELSEM (Tool to Estimate Land-Surface Emissivities at Microwave frequencies) [22], and found that both results agreed reasonably well over snow-free areas, while larger differences occurred over deserts and snow, likely due to the lack of quality inputs for the model on these complex environments. Therefore, there is still a need to better account for the emissivity properties in models under arid and snow environments.

In this work, we focus on evaluating the performance of the CRTM emissivity model by using our ground-based measurements over sand and grass surfaces. Then, we attempt to identify possible problems and make relevant corrections to the model by combining in situ observations.

The structure of this paper is as follows: Section 2 introduces the land emissivity model and the ground-based observation system for multi-surface emissivity. A comparative analysis of these two results is presented in Section 3, while the related discussion is given in Section 4, and finally, a summary of this study is provided in Section 5.

## 2. The Model and Ground-Based Measurements

### 2.1. CRTM Land Emissivity Model

The CRTM model used here is the latest available version V2.4.0 provided by the Joint Center for Satellite Data Assimilation (JCSDA) (https://www.jcsda.org/crtm, accessed on

16 October 2023). The land emissivity model in CRTM was developed by Weng et al. [8], and utilizes scientific advances in various aspects, from atmospheric sciences to electrophysics and astrophysics. Especially, the volumetric scattering theory is adopted to compute the optical parameters of snow, deserts, and canopy leaves. The radiative transfer theory applied in atmosphere is used to compute the bulk-emitted radiation from surface. The roughness effect that is approximated by the small perturbation theory is also introduced in the surface emission and scattering modules. Therefore, the CRTM land emissivity model is able to quantify the emissivity over various surface conditions, including deserts, vegetation, and snow.

The emissivity model is based on the two-stream approximation and adopts different radiation transfer solutions dependent on surface characteristics. The model deals with the radiance transfer processes in three layers: the top layer represents the atmosphere, the middle layer represents the surface cover, and the bottom layer represents the soil. The main treatments in the model are the volume scatterings in the middle layer, and the reflectance at the interfaces between the layers, calculated using the modified Fresnel equations. The detailed process can be found in Weng et al. [8].

The emissivity model framework is shown in Figure 1. As seen from the framework, it consists of two parts, one is the calculations of the dielectric constant [23], reflectance, and reflectance correction on bare soil surface, and then for the canopy surface, the dielectric constant and optical parameters of the vegetation need to be added [24].

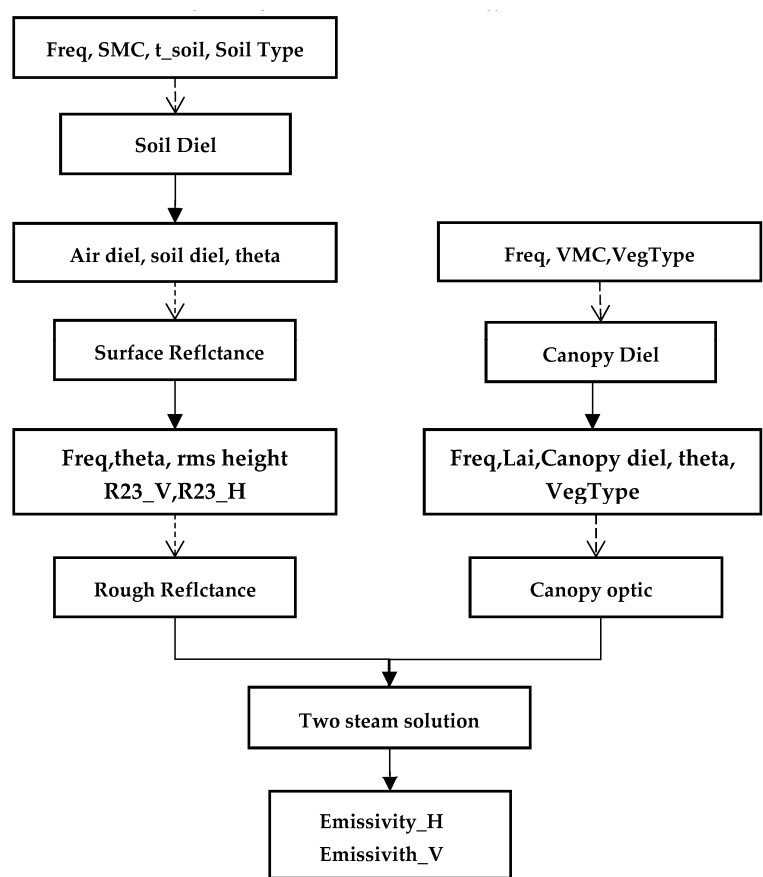

**Figure 1.** The framework of CRTM land emissivity model.

To derive accurate microwave emissivity of the land surface, more detailed surface parameters need to be provided, including the incidence angle, frequency, surface type, soil temperature and moisture, and vegetation characteristics parameters. In this study, the model simulation is mainly compared with the ground-based measurements over grassland and sand surfaces, so the model is divided into vegetation and bare soil (no vegetation)

modules. The input parameters for calculating the surface emissivity over grass and sand are listed in Table 1, especially the key surface variables, such as soil temperature, soil moisture content (SMC), and surface temperature, which can be obtained directly from the ground-based measurements.

**Table 1.** Input parameter configuration in CRTM land emissivity model over grassland and sand.

| Surface | Incidence Angle (°) | Frequency (GHz) | Vegetation Fraction | Soil Type | Vegetation Type | Surface Temperature (°C) | Soil Temperature (°C) | SMC (g/cm³) |
|---------|---------------------|------------------|----------------------|-----------|------------------|---------------------------|------------------------|-------------|
| Sand | 25–65 | 18.7/36.5 | 0 | loamy sand | Bare soil | Measured from Infrared sensor over sand and grass | Measured from probe at 5 cm of soil in sand and grass field | Measured from probe at 5 cm of soil in sand and grass field |
| Grass | | | 0.8 | sandy clay | Short grass | | | |

### 2.2. The Ground-Based Measurements

The ground-based observation data used in this work were collected from the Multi-Surface Observation System at the Xianghe observation site (116.98E, 39.76N), Hebei Province, China. As shown in Figure 2a, the Multi-Surface Observation System mainly includes a mobile platform carrying an RPG microwave radiometer for scanning various typical underlying surfaces, such as pond, cement, grassland, bare soil, and sand surfaces at different angles, along with auxiliary sensors for measuring surface temperature, soil temperature, and soil moisture. As an example, the main observations, including brightness temperature ($Tb$), surface temperature ($Ts$), soil temperature, and SMC at three depths (5 cm, 10 cm, and 15 cm), are shown in Figure 2b–d for the sand surface on 19 March 2020. We can directly see the variation trend of surface radiance at both polarizations with scanning angle over sand, as shown in Figure 2b. Meanwhile, the corresponding surface and soil layer temperatures in Figure 2c display the clear diurnal variations. Relatively, the soil moisture SMC in Figure 2d changed slightly in a day, and the deeper the depth, the smaller the variation. It can be seen that the ground-based microwave radiometer on the mobile platform can scan multiple testing fields at different angles almost simultaneously (less than 1 h), providing high-temporal-resolution in situ measurements to investigate the variation rules of surface emissivity over different surfaces. More details can be found in our previous work [20]. This study focuses on the measurements of sand and grassland surfaces.

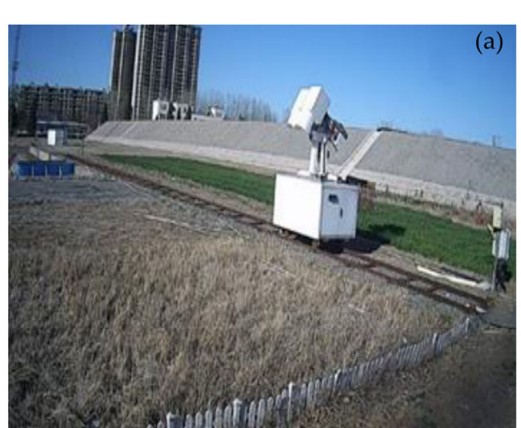
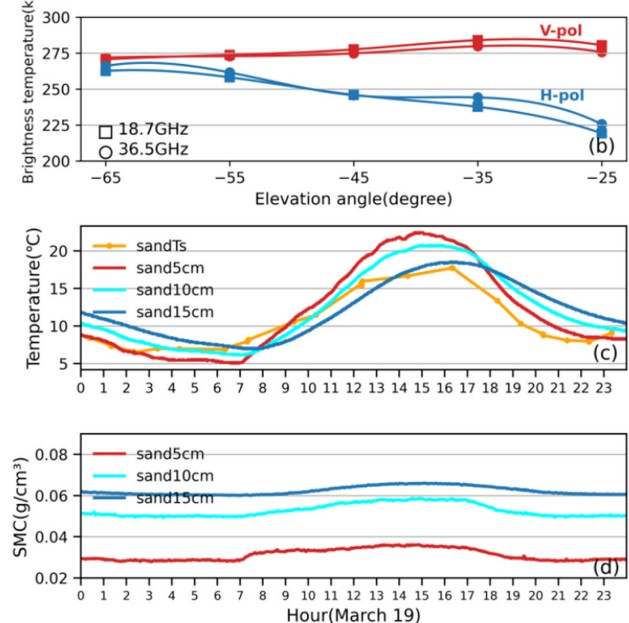

**Figure 2.** The photo of the Multi-Surface Observation System (**a**) and related observations over sand surface on 19 March 2020: (**b**) brightness temperature; (**c**) surface temperature and soil temperature at three layers 5 cm, 10 cm, and 15 cm; (**d**) soil moisture (SMC) at three layers.

When the microwave radiometer is fixed to scan surfaces, the measured Tb reflects the contribution of the upward radiation from the surface and the downward-reflected atmospheric radiation, as shown in Equation (1).

$$T_b = \varepsilon T_s + (1 - \varepsilon) T_{sky} \tag{1}$$

where $\varepsilon$ is the surface microwave emissivity, $T_s$ is the surface temperature, and $T_{sky}$ is the radiation from the sky. By combining the $T_b$ contributions from the sky and the surface measured using the microwave radiometer, along with the synchronous measurement of surface temperature using the infrared thermometer in the observation system, we can calculate the emissivity value using the following Equation (2).

$$\varepsilon = (T_b - T_{sky}) / (T_s - T_{sky}) \tag{2}$$

The microwave radiometer used in this study is the RPG-4CH-DP with dual-frequency (18.7 and 36.5 GHz) and dual-polarization (V-pol and H-pol) [25]. Dual-polarization can detect differences in soil moisture and soil composition by monitoring the polarization signal differences. The two selected frequencies are widely used by satellite-borne microwave imagers, which can provide high-quality measurements for satellite data validation. The infrared thermometer employed in this study was the SI-111 infrared temperature sensor from the Apogee company, used to monitor the surface temperature. The soil temperature and humidity beneath the surface were directly obtained using water content reflectometer, CS655, measured at depths of 5 cm, 10 cm, and 15 cm, respectively. In this work, we mainly selected observation data from 19 to 24 March 2020.

## 3. Results

### 3.1. The Simulations and Measurements over Grass and Sand Surfaces

In this work, we focus on emissivity simulations over the sand and grass surfaces, representing bare and vegetated land surfaces, respectively. For the sand surface, the soil type selected is loamy sand with a sand content of 92% and a clay content of 6%. For the grass surface, the soil type selected is sandy clay with a sand content of 50% and clay content of 43%. The soil temperature and moisture content as well as the surface temperature can be directly obtained by on-site measurements. The microwave radiometer scans each underlying surface from the horizon (0°) to the ground, and the elevation angle is defined as the angle between the scanning direction and the horizontal. Elevation angles ranging from 25° to 65° mainly cover each observed surface.

For the grassland surface, it is necessary to add two parameters, vegetation type and vegetation coverage. Considering the actual growth and vegetation density in the grass field during the selected time range, low vegetation is the more appropriate choice for the vegetation type in the model, with the vegetation coverage rate set at 80%.

Figure 3 shows the comparisons of the simulated and measured surface emissivity over the grass and sand surfaces at an incidence angle of 55°, which is commonly used for satellite microwave observations. The surface emissivity values over grass are mostly high at around 0.95 for both vertical and horizontal polarizations at 18.7 GHz (Figure 3a) and 36.5 GHz (Figure 3b), with the corresponding polarization difference being quite small—approximately 0.02, which varied slightly with time (Figure 3c). The modeled results in dual-frequency dual-polarization are in very good agreement with the measurements. Although slightly lower than the measured values, the simulated emissivity exhibits a more consistent trend and smaller polarization difference, almost identical to the measurements.

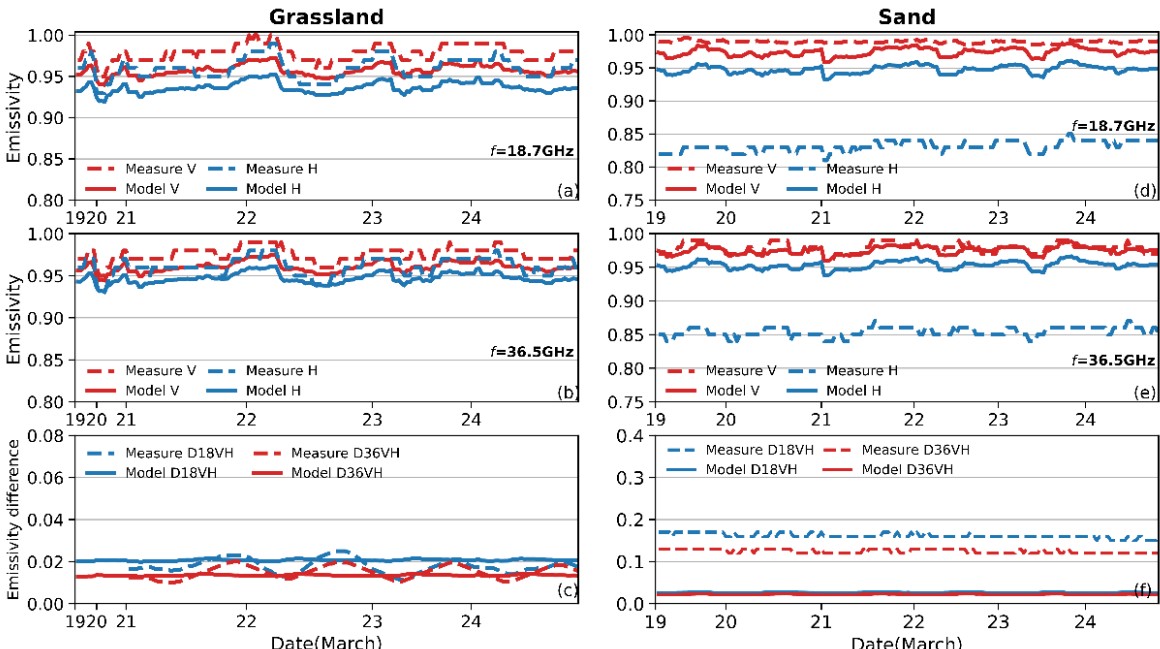

**Figure 3.** Comparison of measured and simulated emissivity in dual-polarization of dual-frequency as well as the corresponding emissivity polarization difference (DVH = V − H) over grassland (**a–c**) and sand (**d–f**) with an incident angle of 55° during 19 to 24 March 2020.

The corresponding comparisons over sand are shown in Figure 3d–f. It is seen that the model simulations are close to the measured emissivity in the vertical polarization. However, a significant difference is observed in the horizontal polarization of both frequencies, as shown in Figure 3d,e for instance, the measured value is about 0.85, while the simulated value is close to 0.95, similar to that over the grass surface. As a result, the large difference in the emissivity polarization difference (DVH = V − H) is seen in Figure 3f. Obviously, the DVH measured at both 18.7 and 36.5 GHz are close to 0.15, and the DVH decreases with increasing frequency. The simulated DVH varies by less than 0.05 and seems independent of the frequency.

The field measurements, with the incident angle ranging from 25° to 65°, mainly cover each observed surface. The corresponding simulated and measured emissivity at those angles are further compared in Figure 4. It is seen that both the simulated and observed emissivity over grassland are close and vary slightly with the angles for dual-polarization and dual-frequency (Figure 4a,b). As a result, most of the emissivity values overlap at each angle, in both horizontal and vertical polarizations. The corresponding emissivity polarization differences are also quite close, constantly less than 0.02 at different angles (Figure 4c).

The measured and modeled emissivity over the sand surface is compared in Figure 4d–f. The model results almost overlapped with the measured emissivity in the vertical polarization at different angles; however, there is a significant difference in the horizontal polarization of both frequencies. The measured emissivity in the horizontal polarization gradually decreases from 0.92 to around 0.75 with increasing incident angle. In contrast, the simulated emissivity shows slight changes with the incident angle, with values mostly above 0.9. This discrepancy leads to a large difference in the emissivity polarization difference (DVH) shown in Figure 4f. For example, the DVH measured at both 18.7 and 36.5 GHz clearly increases to 0.2 as the angle increases to 65°, while the simulated DVH changes slightly within 0–0.05.

The distribution of the probability density function (PDF) for the differences between the simulated and measured emissivity over sand and grassland is compared, as shown in Figure 5. It shows that over grass the peak of PDF for emissivity difference in both

polarizations at the two frequencies is close to zero, and the corresponding MBE values are quite small (~0.01). This indicates that the CRTM land emissivity model with the vegetation module works well for the grassland surface conditions. For the sandy soil surface, the PDF distribution in the vertical polarization is similar to that over grass with the PDF peak around zero, and the corresponding MBEs are less than 0.01. However, the differences in the horizontal polarization are more significant, for instance, the PDF of difference covers a relatively wide range from −0.2 to 0, and the corresponding MBE values are up to 0.08 at both frequencies. The fact that the error bias on the sand surface is several times that on the grass surface has led us to investigate the emissivity model for bare soil in more detail.

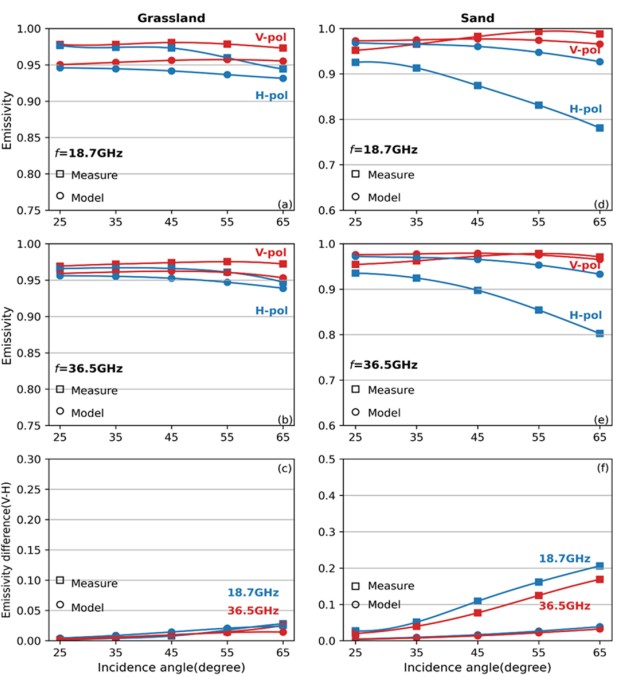

**Figure 4.** The measured and modeled emissivity and polarized difference over grass (**a**–**c**) and sand (**d**–**f**) surface with incident angles ranging from 25° to 65°.

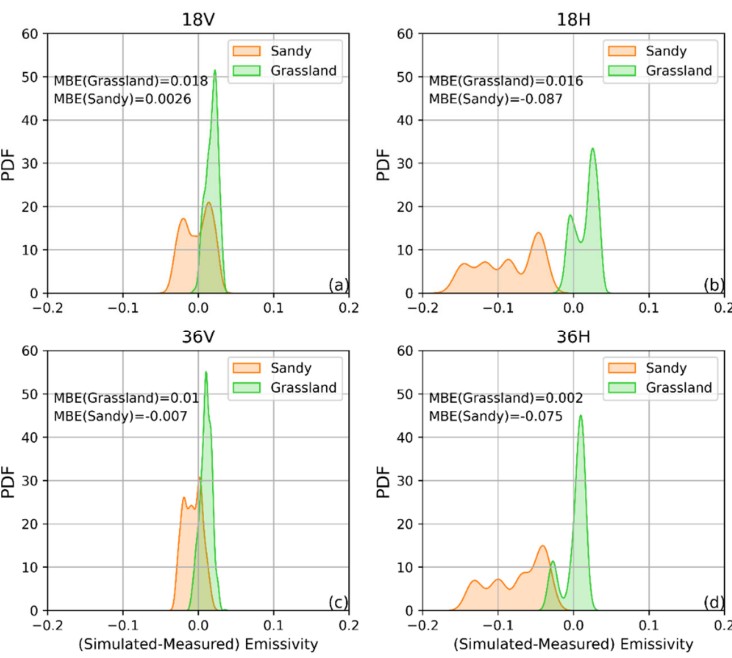

**Figure 5.** The PDF distribution of the differences between simulated and measured emissivity over sandy soil surface and grassland.

### 3.2. Corrections of the Emissivity Model for Bare Soil

The main components in the emissivity model for bare soil include the calculation of the dielectric constant and reflectance and roughness corrections, as shown in the framework in Figure 1. Firstly, we carry out the sensitive tests for the dielectric constant with different input groups, and the corresponding results do not alleviate the issue of overestimated horizontal polarized emissivity, which indicates the dielectric constant is not the main factor in resolving the smaller polarization differences for bare soil. Many previous works have demonstrated that surface roughness significantly affects the reflectance of bare soil [11,26–32]. There are a variety of microwave emissivity models for rough soil surfaces. Then, we focus on the reflectance and roughness correction treatments in the model for the sand surface.

In the current CRTM land emissivity model, the *Q/H* module developed by Choudhury et al. [26] is used to calculate the roughness correction of surface reflectance. The *Q/H* module is currently one of the most commonly used semi-empirical models for describing the effective reflectivity of land surfaces. Generally, there are two roughness factors, *Q* and *H*, to correct the reflectivity in the *Q/H* module. *Q* represents the energy emitted in orthogonal polarization due to the roughness effect on the surface, while H denotes the decrease in effective reflectivity caused by an increase in frequency due to surface roughness. The *Q/H* module we used is defined in Table 2 with *H* = 0.3 [33], and *Q* is a function of frequency, incident angle, and the roughness parameter *s*, where *s* represents the surface root mean square (rms) height. Typically, a value of *s* = 0.25 for a smooth surface, and *s* = 2.15 for a highly rough surface. Given that the sand surface is relatively smooth, we set the *s* = 0.5 in this case.

**Table 2.** The *Q/H* and $Q_p$ module description.

|  | Definition Formula | Relevant Parameters and Settings | Note |
|---|---|---|---|
| *Q/H* | $R_p = H[Q \cdot r_q + (1-Q)r_p]$ | $Q = 0.35(1 - e^{-0.6 \times freq \times \sigma^2})$ <br> Roughness: rms height ($s$) = 0.5 | All surface |
| $Q_p$ | $R_p = Q_p \cdot r_q + (1 - Q_p) \cdot r_p$ | $log[Q_p] = a_p + b_p - log(\frac{s}{l}) + c_p - (s/l)$ <br> Roughness: rms slope ($s/l$) = 0.25 | Bare soil |

Firstly, the change in *Q* and reflectivity in the *Q/H* module with roughness s is shown in Figure 6a. Clearly, *Q* rapidly increases to 0.35 over bare soil with *s* = 0.5, then tends to keep constant over the rough surface. The corresponding polarized reflectivity changes little with surface roughness, remaining mostly below 0.05.

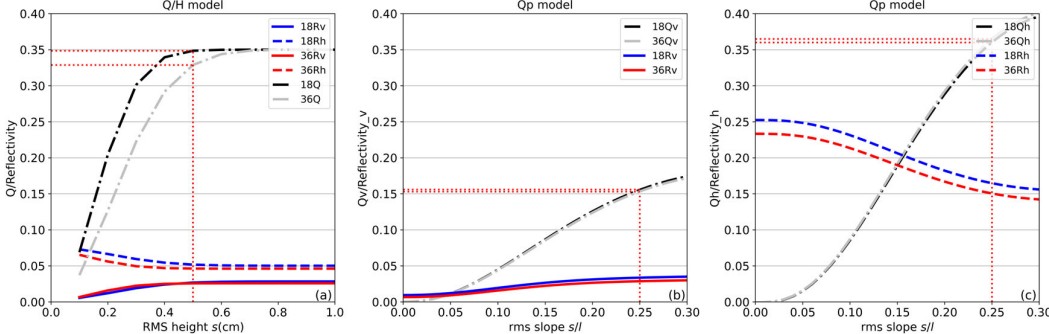

**Figure 6.** The variations in $Q/Q_p$ and the corresponding reflectivity $R_p$ as functions of the roughness parameter s in the *Q/H* module (**a**) and rms slope $s/l$ in the $Q_p$ module ((**b**,**c**), for V and H polarizations, respectively) at 18.7 GHz and 36.5 GHz with incident angle of 55°. The red dotted line represents the calculated value when using the default setting of $s = 0.5$ (in (**a**)) and $s/l = 0.25$ (**b**,**c**).

It should be noted that the single *Q* in the *Q/H* module is employed to describe the impact of surface roughness on polarized reflectivity, suggesting that the surface roughness

effect on reflectance is independent of polarization. This treatment in the *Q/H* module results in almost identical emissivity in both vertical and horizontal polarization, which differs significantly from the $I^2EM$ model simulations mentioned in Shi et al. [27] and the measurements over the sand surface as shown in Figure 3.

To address this issue, we introduce the $Q_p$ module developed by Shi et al. [27] which combines emissivity measured during the PORTOS-93 field campaign and simulations from $I^2EM$. The $Q_p$ module is a multi-frequency and multi-polarization land surface reflectance module for bare soil, referred to as a modification of the *Q/H* module.

As the definition of the $Q_p$ module in Table 2, the effect of surface roughness on reflectance is mainly characterized by $Q_p$, and the subscript $p$ denotes the polarization state. Hence, using the polarized $Q_p$ is able to correct the roughness effect in different polarizations for bare soil. $Q_p$ is dependent on both the frequency and the roughness parameter, rms slope $s/l$, which is the ratio of the surface rms height s to correlation length $l$. Initially, $Q_p$ was specifically developed for 10.7 GHz, and then, $Q_p$(f) expressions in Table 2 were derived for more frequencies. A more detailed description can be found in Shi et al. [27].

Differing from using the single $Q$ for both polarizations in the *Q/H* module, the $Q_p$ module adopts the polarized $Q_v$ and $Q_h$ based on nonlinear functions of the $s/l$ value. Here we have set $s/l$ to 0.25 for the relatively smooth sand surface. The variation in $Q_p$ and reflectivity with the roughness parameter $s/l$ at both frequencies are shown in Figure 6b,c.

Clearly, the increasing rate of $Q_h$ looks similar to $Q$ in the *Q/H* module, and $Q_v$ appears to grow more slowly than $Q_h$ over bare soil. The corresponding reflectivity $R_h$ decreases gradually to 0.15 when the $s/l$ equals 0.25, and $R_v$ varies slightly, mostly remaining within 0.05. Compared with the results in Figure 6a, both $R_v$ values in the *Q/H* and $Q_p$ module are close and small but $R_h$ in the $Q_p$ module is more than three times larger than that in the *Q/H* module.

Subsequently, Figure 7a,b presents the comparisons between the measured and simulated emissivity using the $Q_p$ module at two frequencies for an incident angle of 55° during 19 to 24 March 2020. Clearly, the simulations in the horizontal polarization almost perfectly overlapped with the measurements at both frequencies, with even better alignment at 18.7 GHz. The corresponding emissivity polarization differences shown in Figure 7c are also quite minimal, clearly differing from the results shown in Figure 3 for the sand surface. The high consistency between the simulated and measured emissivity over the sand surface suggests that the $Q_p$ module can significantly improve the accuracy of the simulation, especially in the horizontal polarization; however, minor differences are observed in vertical polarization at 18.7 GHz.

Figure 7d–f further shows the updated emissivity simulations at various incidence angles using the $Q_p$ module against the corresponding measurements. It is evident that the simulations for horizontal polarization of 18.7 and 36.5 GHz agree well with the measurements at different angles, as shown in Figure 7d,e; however, the simulations for vertical polarization tend to be slightly lower. The overall trend of polarization differences between the simulated and measured results at both frequencies is very close and consistent in Figure 7f, highlighting the significant improvement in horizontal polarization obtained by the $Q_p$ module.

The boxplot of emissivity differences between the measurements and simulations using both *Q/H* and $Q_p$ modules over sand are compared and presented in Figure 8. It is evident that the medians of emissivity difference in the vertical polarizations at both frequencies are close and smaller than 0.03. This indicates that both *Q/H* and $Q_p$ modules perform well in the vertical polarization for bare soil; however, the differences in horizontal polarizations are more significant. For instance, using the $Q_p$ module covers a narrow range around zero, and the corresponding MBEs are significantly reduced from 0.08 in the *Q/H* module to less than 0.02 at both frequencies. These comparisons further proved that for bare soil surfaces, such as sand that exhibits significant polarization differences, using

the $Q_p$ module to correct the roughness effect on reflectivity is an optimal selection. This approach can achieve notable improvements in horizontal polarization.

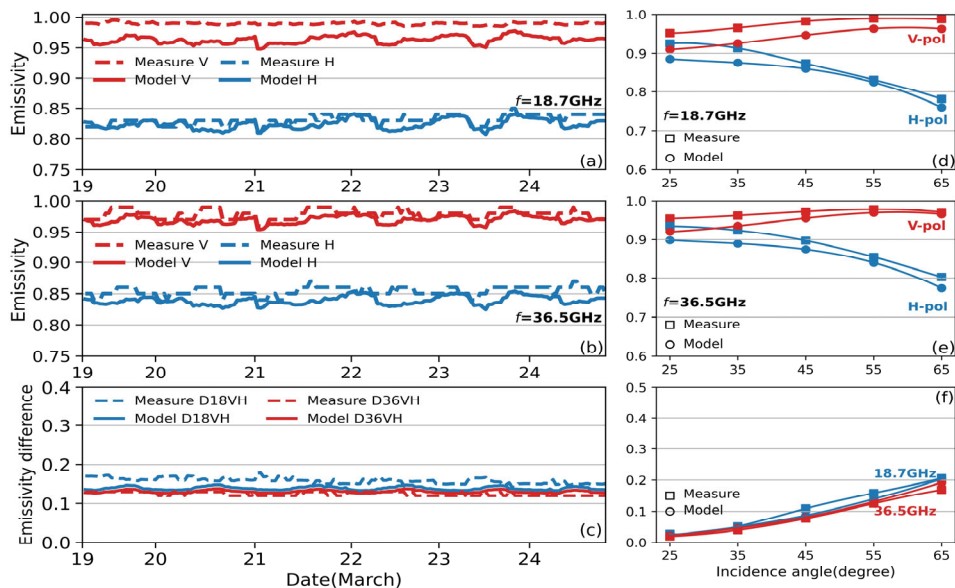

**Figure 7.** Comparison of emissivity simulations using $Q_p$ module and measurements as well as polarization difference over sand surface with an incident angle of 55° (**a–c**) and with incident angles ranging from 25° to 65° (**d–f**) during 19 to 24 March 2020.

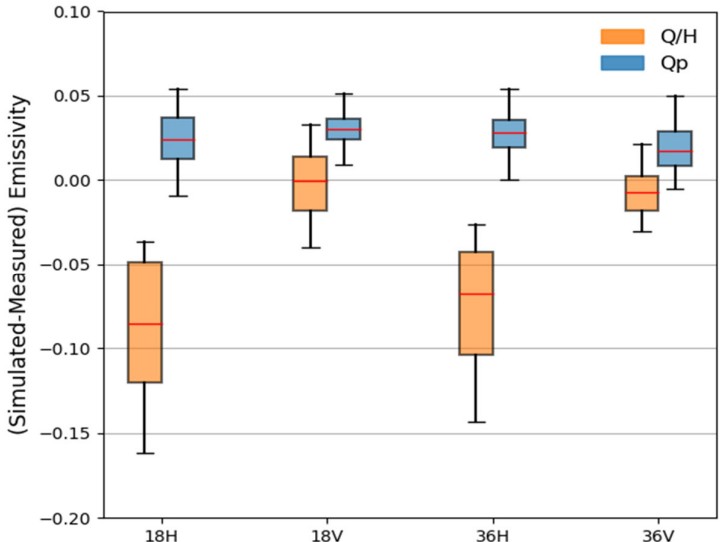

**Figure 8.** The boxplot of the emissivity differences between measurements and simulations using both *Q/H* and $Q_p$ modules over the sand surface.

## 4. Discussion

The ground-based measurements reflect that there is a significant difference in the emissivity between horizontal and vertical polarization when the underlying surface is a relatively smooth bare soil. For instance, the polarization difference over the sand surface can reach about 0.2 when the incident angle is 65°, while for the grassland surface, due to the multiple scattering effects of vegetation leaves, the polarization differences in surface emissivity are significantly weakened. Therefore, the measured surface emissivity in vertical and horizontal polarization is very close over grassland, and the polarization difference is smaller than 0.02 and stable at different incident angles.

Combined with the treatments of surface roughness on reflectance, the *Q/H* module assumes that the roughness effect on both the vertical and horizontal polarizations is identical, so the effect of surface roughness on emissivity is independent of polarization. For the *Q/H* module used in the CRTM land emissivity model, the parameter *H* is set to 0.3 and the *Q* value depends only on the surface rms height and frequency. In contrast, the $Q_p$ module takes account of different parameters to deal with the roughness effect on the vertical and horizontal polarizations. Therefore, for the emissivity of bare soils (e.g., sand surface) with significant polarization differences, using the *Q/H* module could lead to more inaccuracies, while the $Q_p$ module is the optimal option because it can significantly improve the accuracy of emissivity in horizontal polarization. On the other hand, for the emissivity of vegetation surfaces (e.g., grass surface) with slight polarization differences, using the *Q/H* module in the emissivity model demonstrates good simulation results, showing its suitability for vegetated land surface.

In addition, the Multi-Surface Observation System can provide in situ measurements for more surfaces, such as cement, bare soil, and ponds. We also need to further combine the emissivity model with the measurements on these different underlying surfaces to investigate the feasibility of detailed treatments for the dielectric constant, reflectance, and reflectance roughness correction on those underlying surfaces.

## 5. Conclusions

This study utilized ground measurement data from the multi-surface emissivity observation system at the Xianghe site to evaluate the CRTM land emissivity model covering various frequencies, incident angles, and surface types. The measured surface temperature as well as soil temperature and moisture are used as real input parameters for the emissivity model. The model simulations are then compared with on-site emissivity measurements from dual-frequency and dual-polarized microwave radiometer observations over grassland and sand surfaces. The results show that the emissivity simulation and ground measurements agree quite well at both polarizations over the grassland surface but exhibit a more significant difference in the horizontal polarization over the sand surface, i.e., with MBE about $-0.087$ at 18.7 GHz and $-0.075$ at 36.5 GHz. To address this issue, the *Q/H* module used in the CRTM emissivity model for soil reflectivity roughness correction was replaced with the $Q_p$ module, resulting in a significant reduction in MBE for horizontal polarization at both frequencies (i.e., 0.024 at 18.7 GHz and 0.027 at 36.5 GHz). These results indicate that the $Q_p$ module can significantly improve the roughness effect on emissivity at horizontal polarizations over sand surfaces. For grassland, the CRTM emissivity model with the *Q/H* module demonstrates good simulations, showing its suitability for vegetated land surfaces.

The model simulations for sand and grassland are compared with ground measurements within a few days in this paper. More in situ measurements covering various seasons and years are still needed to assess and improve model performance. It should also be noted that there are some uncertainties existing in the ground measurements, such as potentially introducing measurement errors by external factors. Further studies should conduct a comprehensive examination and reduce these uncertainties as much as possible.

**Author Contributions:** Data curation, W.N.; Writing—original draft, Y.W. and W.H.; Writing—review & editing, M.D., H.L. and C.H.; Project administration, H.C. All authors have read and agreed to the published version of the manuscript.

**Funding:** This work was supported by the National Key Research and Development Program of China [2022YFF0801301] and the National Natural Science Foundation of China [No. 41575033], the Instrument Function Development Project of the Chinese Academy of Sciences [No. E066193601], and the Natural Science Foundation of Jiangsu Province [BK20231148].

**Data Availability Statement:** Not applicable.

**Acknowledgments:** We thank the staff at the Xianghe site for their maintenance work on the microwave radiometer and the ground mobile observation system, especially Qun Cheng, Qing Yao. We thank JCSDA for providing the CRTM model for free download.

**Conflicts of Interest:** The authors declare no conflict of interest.

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
