# Peer review of "Evaluation of the CRTM Land Emissivity Model over Grass and Sand Surfaces Using Ground-Based Measurements"

_remotesensing, doi:10.3390/rs16010095_

Round 1

Reviewer 1 Report

Comments and Suggestions for Authors

This study simulates the land surface microwave emissivity by combining the CRTM model with ground measurements. The length of the article is too short as the scientific paper of the Remote Sensing periodical, and the content is also brief. The title covers a wide range, and does not offer the special work of the study. The following is the specific suggestions.

Major issues:

1. There is no analysis for Figures 7(c), (d) and (e).

2. From Figure 8, the Qp module more effectively corrects the roughness effect in horizontal polarization, but not in the vertical polarization. Therefore, it is not appropriate in the Abstract section about the expression “The adjustment demonstrates that the Qp module more effectively corrects the roughness effect on reflectivity for bare soil surfaces.”

3. Why does a significant difference appear in the horizontal polarization in Figures 3 (d), (e) and (f)? Please give the explanation for this phenomenon. The authors think that the modeled results closely match the measurements at both frequencies from Figures 3 (a), (b) and (c). The matching effect in Figures 3 (a) and (b) is good, but not Figure 3 (c)

Minor issues:

1. Section 1, there is no references in Paragraph 2.

2. There is no Table 1 in the whole article. The texts in Figure 2 (a) are blurry, and Figure 2 (b) is a table, not a figure.

Comments on the Quality of English Language

Minor editing of English language required

Author Response

Comments and Suggestions for Authors

This study simulates the land surface microwave emissivity by combining the CRTM model with ground measurements. The length of the article is too short as the scientific paper of the Remote Sensing periodical, and the content is also brief. The title covers a wide range, and does not offer the special work of the study.

Reply: Thanks for your comments and suggestions. We fully agree with your suggestions and have adjusted the main objective of the manuscript, which is to evaluate and improve the CRTM land emissivity model by combining our ground-based measurements. Therefore, 1) the title has been changed to “Evaluation of CRTM land emissivity model over grass and sand surface by using ground-based measurements”.  2) About the length of the article, currently it is about 4500 words in the revised manuscript, which meets the length requirements of article.

The following is the specific suggestions.

Major issues.

  1. There is no analysis for Figures 7(c),(d) and (e).

Reply: Sorry for this mistake. We added the related description in Page 11, seen “Figure 7(d, e, f) further shows the updated emissivity simulations at various incidence angles using the Qp module against the corresponding measurements. It is evident that the simulations for horizontal polarization of 18.7 and 36.5 GHz agree well with the measurements in different angles, as shown in Figure 7 (d, e). However, the simulations for vertical polarization tend to be slightly lower. The overall trend of polarization differences between the simulated and measured results at both frequencies is very close and consistent in Figure 7 f, highlighting the significant improvement in horizontal polarization obtained by the Qp module.”

  1. From Figure 8, the Qp module more effectively corrects the roughness effect in horizontal polarization, but not in the vertical polarization. Therefore, it is not appropriate in the Abstract section about the expression “The adjustment demonstrates that the Qp module more effectively corrects the roughness effect on reflectivity for bare soil surfaces.

Reply: Thank you. We agree with your comments, and changed this sentence in Abstract into “The adjustment demonstrates that the Qp module more effectively corrects the roughness effect on horizontally polarized emissivity for bare soil surfaces”.

  1. Why does a significant difference appear in the horizontal polarization in Figures 3 (d), (e) and(f)? Please give the explanation for this phenomenon. The authors think that the modeled results closelv match the measurements at both frequencies from Figures 3 (a). (b) and (c). The matching effect in Figures 3 (a) and (b) is good, but not Figure 3 (c).

Reply: Thanks for your questions. Sorry for the confusing descriptions in the paper about Figure 3. The results shown in Figure 3 reflect the difference of surface microwave emissivity between simulations derived from CRTM land emissivity model and ground-based measurements over grassland (Figure 3(a, b, and c)) and sand (Figure 3(d, e, and f)), especially the Figure 3c and Figure 3f are the corresponding polarized difference (DVH= V-H) of emissivity at both frequencies. Clearly, over grassland, the model results are close to the measurements at both frequencies as shown in Figure 3 (a, b), so that their corresponding polarized difference of emissivity are relatively smaller in Figure 3c.

Over sand, the model simulations at the horizonal polarizations are much higher than the corresponding measured emissivity as shown in Figure 3(d, e) for 18.7 and 36.5 GHz, respectively, so their corresponding polarized difference shown in Figure 3f are quite larger, such as about 0.15 for measurement while about 0.02 for model simulations.

About why a significant difference appearing in the horizonal polarization over sand, we conducted more investigations in the paper, and found that the reason is mainly related to the treatment of reflectance and roughness corrections in the current CRTM land emissivity model for bare soil.

Therefore, the corresponding modifications are seen in page 6 “Figure 3 shows the comparisons of the simulated and measured surface emissivity over grass and sand surface at an incidence angle of 55° which is commonly used for satellite microwave observations. The surface emissivity values over grass are mostly high at around 0.95 for both vertical and horizontal polarizations at 18.7 GHz (Figure 3 a) and 36.5 GHz (Figure 3 b), with the corresponding polarization difference being quite small (approximately 0.02 and varied slightly with time (Figure 3 c). The modeled results in dual-frequency dual-polarization are in very good agreement with the measurements. Although slightly lower than the measured values, the simulated emissivity exhibits a more consistent trend and smaller polarization difference, almost identical to the measurements.

The corresponding comparisons over sand are shown in Figure3 (d, e, f). It is seen that the model simulations are close to the measured emissivity in the vertical polarization. However, a significant difference is observed in the horizontal polarization of both frequencies as shown in Figure 3(d, e), for instance, the measured value is about 0.85, while the simulated value is close to 0.95, similar to that over grass surface. As a result, the large difference in the emissivity polarization difference (DVH=V-H) is seen in Figure 3f. Obviously, the DVH measured at both 18.7 and 36.5 GHz are close to 0.15, and the DVH decreases with increasing frequency. The simulated DVH varies by less than 0.05, and seems independent of the frequency.

Figure 3. Comparison of measured and simulated emissivity as well as the corresponding emissivity polarization difference (DVH=V-H) over grassland (a, b, c) and sand (d, e, f) with an incident angle of 55°during March 19 to 24, 2020.

Minor issues:

  1. Section 1, there is no references in Paragraph 2

Reply: Thank you. We added the references in Paragraph 2 as “Field experiments under controlled conditions can provide high temporal resolution of surface emissivity data, allowing detailed analysis of the impact of surface processes on emissivity [5-7] .”

Then they were listed in references,

5   Wigneron, J.-P.; Laguerre, L.; Kerr, Y. A simple parameterization of the L-band microwave emission from rough agricultural soils. Geoscience and Remote Sensing, IEEE Transactions on 2001, 39, 1697-1707, doi:10.1109/36.942548.

  1. Wang, J.R.; Neill, P.E.O.; Jackson, T.J.; Engman, E.T. Multifrequency Measurements of the Effects of Soil Moisture, Soil Texture, And Surface Roughness. IEEE Trans. Geosci. Remote Sens. 1983, GE-21, 44-51, doi:10.1109/TGRS.1983.350529.
  2. Coppo, P.; Luzi, G.; Paloscia, S.; Pampaloni, P. Effect of soil roughness on microwave emission: comparison between experimental data and models. In Proceedings of the [Proceedings] IGARSS'91 Remote Sensing: Global Monitoring for Earth Management, 3-6 June 1991, 1991; pp. 1167-1170.

  1. There is no Table 1 in the whole article. The texts in Figure 2 (a) are blurry, and Figure 2 (b) is a table, not a figure.

Reply: Sorry for the mistake. We added the Table 1 in Page 4 and the related descriptions, such as “The input parameters for calculating the surface emissivity over grass and sand are listed in Table 1, especially the key surface variables, such as soil temperature, soil moisture content (SMC), and surface temperature, can be obtained directly from the ground-based measurements.”.

  We also adjusted the text clearly in Figure 1 (formerly Figure 2(a)) shown in Page 3. Both of them are shown here.

Figure 1. The framework of CRTM land emissivity model

Table 1. Input parameters configuration in CRTM land emissivity model over grassland and sand

Surface

Incidence

angle(°)

Frequence

(GHz)

Vegetation

Fraction

Soil type

Vegetation type

Surface Temperature()

Soil

Temperature()

SMC

(g/cm3)

Sand

25-65

18.7/36.5

0

loamy sand

Bare soil

Measured from Infrared sensor

over sand and grass

Measured from probe at 5cm of soil in sand and grass field

Measured from probe at 5cm of soil in sand and grass field

Grass

0.8

sandy clay

Short grass

Reviewer 2 Report

Comments and Suggestions for Authors

The manuscript compares the simulated CRTM measurements of surface microwave emissions with the ground-based measurements over a grassland and a sandy terrain. I have the following comments for the authors:

1. The main focus of the paper should not be the comparison between ground-based and simulated data of microwave surface emissions from different acquisition modes and surface conditions, as stated in the manuscript. This is a weak scientific question to be emphasized in the manuscript. Instead, the main objective should be the improvement of the CRTM estimations by a new method of correcting soil reflectivity due to roughness in the CRTM model. Thus, title, objective, and main conclusion should be changed to fit this new approach.

2. The structure of the paper can be improved in order to make the reading smoother. My suggestion is as follow:

1. Introduction

2. CRTM surface emissivity model (please, give proper credit to the authors who proposed this model).

3. Methods

3.1. Study area

3.2. Data gathering (please, add more details about how the simulated data were obtained)

3.3. Data analysis (several details of methods are presented in the section of the Results)

4. Results

5. Discussion

6. Conclusions

3. Please, identify the meanings of the CRTM, Q/H, and QP in the Abstract. I also would like to ask the authors to double check if all short names in the text are identified at the first time they are reported.

4. The citations are a bit old. Among the 25 citations, the most recent is from 2021. I suggest adding a couple of more recent citations to the manuscript.

5. All figures are nicely prepared. Congratulations.

Comments on the Quality of English Language

Minor editing is demanded.

Author Response

Comments and Suggestions for Authors

The manuscript compares the simulated CRTM measurements of surface microwave emissions with the ground-based measurements over a grassland and a sandy terrain.  have the following comments for the authors.

  1. The main focus of the paper should not be the comparison between ground-based and simulated data of microwave surface emissions from different acquisition modes and surface conditions, as stated in the manuscript. This is a weak scientific question to be emphasized in the manuscript. Instead, the main objective should be the improvement of the CRTM estimations by a new method of correcting soil reflectivity due to roughness in the CRTM model. Thus, title objective and main conclusion should be changed to fit this new approach.

Reply: Thanks for your valuable comments and suggestions. We fully agree with your suggestions and have adjusted the main objective of the manuscript, which is focus on evaluating and improving the CRTM land emissivity model by combining our ground-based measurements. Therefore, the title has been changed to “Evaluation of CRTM land emissivity model over grass and sand surface by using ground-based measurements”.

The corresponding modifications have done in Abstract, such as “The Community Radiative Transfer Model (CRTM) land emissivity model is a useful tool for providing microwave emissivity over complex surface. By combining with ground measurements from a mobile multi-surface observing system at Xianghe site, China, the performance of the land emissivity model is evaluated over grassland and sand surface.”, and in the Introduction part, such as “In this work, we focus on evaluating the performance of CRTM emissivity model by using our ground-based measurements over sand and grass surface. Then we attempt to identify possible problems and make relevant corrections to the model by combing in-situ observations.” in page 2

  1. The structure of the paper can be improved in order to make the reading smoother. My suggestion is as follow.

1.Introduction

  1. CRTM surface emissivity model (please, give proper credit to the authors who proposed this model).

3.Methods

3.1.Study area

  1. Data gathering (please. add more details about how the simulated data were obtained3.3. Data analysis (several details of methods are presented in the section of the Results)

4.Results

5.Discussion

6.Conclusions

Reply: Thanks for your thoughtful suggestions.

Based on the change of title, firstly, the main structure adjustments have done in the section 2. It has changed to

“2. The model and ground-based measurements

2.1 CRTM land emissivity model

2.2. The ground-based measurements”

Then, the more related descriptions about CRTM land emissivity model was added in 2.1, such as “The CRTM model used here is the latest available version V2.4.0 provided by the Joint Center for Satellite Data Assimilation (JCSDA) (https://www.jcsda.org/crtm). The land emissivity model in CRTM is developed by Weng et al. [8], and utilizes the scientific advances in various aspects from atmospheric sciences to electrophysics and astrophysics. Especially, the volumetric scattering theory is adopted to compute the optical parameters of snow, deserts and canopy leaves. The radiative transfer theory applied in atmosphere is used to compute the bulk-emitted radiation from surface. The roughness effect that is approximated by the small perturbation theory is also introduced in the surface emission and scattering modules. Therefore, the CRTM land emissivity model is able to quantify the emissivity over various surface conditions, including deserts, vegetation and snow. ”

In the Introduction, we also added related work about model evaluations in red color, such as “The land emissivity model developed by Weng et al. [8] has been widely used in the Community Radiative Transfer Model (CRTM) for radiance assimilation in numerical weather prediction models. Prigent et al. [21] evaluated and compared the CRTM emissivity model with the global land surface emissivity calculated by TELSEM (Tool to Estimate Land Surface Emissivities at Microwaves) [22], and found that both results agreed reasonably well over snow-free areas, while larger differences occurred over deserts and snow, likely due to the lack of quality inputs to the model in these complex environments. Therefore, it still need take better account of the emissivity properties in models under arid and snow environments.

In this work, we focus on evaluating the performance of CRTM emissivity model by using our ground-based measurements over sand and grass surface. Then we attempt to identify possible problems and make relevant corrections to the model by combing in-situ observations. “

The structure of this paper is as follows: Section 2 introduces the land emissivity model and the ground-based observation system for multi-surface emissivity. “

Considering the overall structure of the paper as well as the emphasis on the process of problem analysis and resolution, it is better to retain the structure and content of the other parts except for the modifications mentioned above. Thanks for your understanding.

  1. Please. identify the meaning as of the CRTM. Q/H. and Qp in the Abstract l also would like to ask the authors to double check if all short names in the text are identified at the first time they are reported.

Reply: Thank you. We modified the short names CRTM in Abstract as “The Community Radiative Transfer Model (CRTM) land emissivity model is a useful tool for providing microwave emissivity over complex surface.”

As to Q/H and Qp, they were named and shown in the same way in most of previous related references.

For other short names in the text, such as JCSDA, it was added full name in the first time “The CRTM model used here is the latest available version V2.4.0 provided by the Joint Center for Satellite Data Assimilation (JCSDA).”

  1. The citations are a bit old. Among the 25 citations, the most recent is from 2021.I suggest adding a couple of more recent citations to the manuscript.

Reply: Thank you. We have added a couple of more recent citations to the manuscript, such as references 17,18,19 published in 2022 and 2021.

  1. Wang, X.; Wang, Z. Microwave Emissivity of Typical Vegetated Land Types Based on AMSR2. Remote Sens. 2022, 14, doi:10.3390/rs14174276.
  2. Li, R.; Hu, J.; Wu, S.; Zhang, P.; Letu, H.; Wang, Y.; Wang, X.; Fu, Y.; Zhou, R.; Sun, L. Spatiotemporal Variations of Microwave Land Surface Emissivity (MLSE) over China Derived from Four-Year Recalibrated Fengyun 3B MWRI Data. Advances in Atmospheric Sciences 2022, 39, doi:10.1007/s00376-022-1314-0.
  3. Hu, J.; Fu, Y.; Zhang, P.; Min, Q.; Gao, Z.; Wu, S.; Li, R. remote sensing Satellite Retrieval of Microwave Land Surface Emissivity under Clear and Cloudy Skies in China Using Observations from AMSR-E and MODIS. Remote Sens. 2021, 13, doi:10.3390/rs13193980.

5.All figures are nicely prepared. Congratulations.

Reply: Thank you for your encouragement.

Reviewer 3 Report

Comments and Suggestions for Authors

An interesting tour of measuring LSME. The graphics are clear enough for the average reader to understand and follow.

Comments on the Quality of English Language

Overall the quality of the English appears acceptable. A few minor changes are needed. For example, in the abstract:

The simulated and measured emissivity are quite agrees at both polarizations over the grassland surfaces, but a more significant difference is observed at the horizontal polarization over sand surface.

Author Response

Comments and suggestions for Authors

An interesting tour of measuring LSME. The graphics are clear enough for the average reader to understand and follow.

Reply: Thank you for your encouragement.

Comments on the Language

Overall the quality of the English appears acceptable. A few minor changes are needed. For example.in the abstract:

The simulated and measured emissivity are agrees at both polarizations over the grassland surfaces. but a more significant difference is observed at the horizontal polarization over sand surface.

Reply: Thank you. We modified it into “The simulated and measured emissivity agrees very well at both polarizations over the grassland surfaces, but a more significant difference is observed at the horizontal polarization over sand surface.”

Round 2

Reviewer 1 Report

Comments and Suggestions for Authors The manuscript has been modified according to my suggestions.

Reviewer 2 Report

Comments and Suggestions for Authors

All my previous concerns were addressed properly. I consider this revised version of the manuscript ready to be published.

Comments on the Quality of English Language

Only minor editing is required.